# Heat Emergencies: Perceptions and Practices of Community Members and Emergency Department Healthcare Providers in Karachi, Pakistan: A Qualitative Study

**DOI:** 10.3390/ijerph18094736

**Published:** 2021-04-29

**Authors:** Uzma Rahim Khan, Naveed Ahmed, Rubaba Naeem, Umerdad Khudadad, Sarwat Masud, Nadeem Ullah Khan, Junaid Abdul Razzak

**Affiliations:** 1Department of Emergency Medicine, Aga Khan University, Karachi 74800, Pakistan; uzma.khan@aku.edu (U.R.K.); naveed.ahmad@aku.edu (N.A.); rubabakhan25@gmail.com (R.N.); umerhaideri47@gmail.com (U.K.); sarwat.masud@scholar.aku.edu (S.M.); nadeemullah.khan@aku.edu (N.U.K.); 2Centre for Global Emergency Care, Department of Emergency Medicine, Johns Hopkins University School of Medicine, Baltimore, MD 21218, USA; 3Centre of Excellence for Trauma and Emergencies, Aga Khan University, Karachi 74800, Pakistan

**Keywords:** heat emergencies, heat exposure, extreme heat events, perceptions, Pakistan

## Abstract

Heat waves are the second leading cause of weather-related morbidity and mortality affecting millions of individuals globally, every year. The aim of this study was to understand the perceptions and practices of community residents and healthcare professionals with respect to identification and treatment of heat emergencies. A qualitative study was conducted using focus group discussions and in-depth interviews, with the residents of an urban squatter settlement, community health workers, and physicians and nurses working in the emergency departments of three local hospitals in Karachi. Data was analyzed using content analysis. The themes that emerged were (1) perceptions of the community on heat emergencies; (2) recognition and early treatment at home; (3) access and quality of care in the hospital; (4) recognition and treatment at the health facility; (5) facility level plan; (6) training. Community members were able to recognize dehydration as a heat emergency. Males, elderly, and school-going children were considered at high risk for heat emergencies. The timely treatment of heat emergencies was widely linked with availability of financial resources. Limited availability of water, electricity, and open public spaces were identified as risk factors for heat emergencies. Home based remedies were reported as the preferred practice for treatment by community members. Both community members and healthcare professionals were cognizant of recognizing heat related emergencies.

## 1. Introduction

Heat emergencies are a public health problem of significance. Global climate change has been related to rising temperatures, which have resulted in heatwaves, more severe droughts, heavy rains, and intense hurricanes in various parts of the world [1,2]. These heat related emergencies have affected communities that were unprepared to handle them. Extreme air temperature contributes directly to cardiovascular and respiratory disease deaths, especially among the elderly. More than 166,000 people died as a result of heatwaves between 1998 and 2017, including more than 70,000 in Europe during the 2003 heatwave. Between 2000 and 2016, the number of people exposed to heatwaves went up by nearly 125 million.

The UNHCR’s report on weather-related incidents from 1995 to 2015 highlighted the importance of risk identification and reduction in vulnerable countries. [3]. With regard to heat events, Chicago, United States reported 514 deaths in July 1995 [4] followed by California that recorded a higher number of emergency visits in 2006 [5]. Similar burden of heat emergencies has been observed in Western Europe including France, Germany and Spain [6,7]. In addition, New South Wales reported a 13% rise in all-cause mortality, and a 14% increase in ambulance calls related to heat emergencies in 2011 [8]. Furthermore, India reported 1000 excessive deaths in Ahmedabad as a result of extreme heat in 2010 [9].

Pakistan ranks 7th on the list of the top 10 countries with the climate risk index score of 30.5 and a total number of 141 weather-related events in the past 10 years [10]. Karachi, being located in a hot climate area is at risk for heat related emergencies, due to the high concentration of infrastructure and limited greenery [11]. Temperatures soar to an extreme of 45 °C during the months of May to July, along with high humidity of 30–95% as a daily average from 2006 to 2015 [12]. The 30-year average maximum temperature for June, the hottest summer month in Karachi, is 34.8 °C [13]. There was a “heat spell” in Karachi in June 2015, causing a state of emergency in the city, claiming more than 1200 lives in a span of four days [14].

Heat emergencies include dehydration, heat cramps, heat exhaustion, heat stroke, and death [15]. Quantitative literature about the epidemiology of heat emergencies has identified that the elderly, children, outdoor workers, and people with comorbidities are at greater risk for heat emergencies [1]. The approach used by most of the epidemiologists to measure heat illnesses has been quantitative [16,17]. Some researchers have identified the need for analyzing the vulnerability of at-risk populations through a qualitative approach. One such study from Sweden explored communities’ perceptions regarding city heat and identified social isolation and female gender as risk factors for heat emergencies in addition to other known risks [16]. A similar study from Australia identified that negative perceptions about heat disasters among the elderly increased their risk for heat illness and hospitalization [18]. Some qualitative studies have explored the perceptions of outdoor workers towards occupational heat injuries, but little is known about the perceptions of healthcare providers and vulnerable communities towards heat emergencies [15,19,20,21,22,23,24].

It is hypothesized that a lack of water supply, long and unpredictable power outages, and a lack of awareness among communities and residents contributed to the high number of heat-related deaths in Karachi in 2015, but evidence is lacking [12]. To better understand the threats that heat-related emergencies pose to Karachi’s populations, the perceptions of local community members and healthcare workers must be explored, the findings of which can be used to develop heat-related emergency prevention strategies [25]. Therefore, the aim of this study was to understand the perceptions and practices of healthcare providers, along with community members regarding diagnosing and treating heat emergencies in Karachi.

## 2. Materials and Methods

### 2.1. Study Design

The study was conducted in November 2017, in Karachi, Pakistan, as part of the Heat Emergency Awareness and Treatment (HEAT) trial (NCT03513315). The HEAT trial was carried out in the months of summer. However, this qualitative study was embedded in the HEAT trial and conducted in November considering that heatwave is a tropical phenomenon in Pakistan. In Karachi, the temperature in the month of November remains moderate ranging from 20 to 32 °C. In addition, we think that the perception about heat is independent of weather conditions. One of the objectives in the heat trial was to develop a community heat prevention guideline. To facilitate the contextual development of the guideline catering to the needs of community and emergency healthcare workers, a qualitative study was conducted using focus group conversations (FGDs) and in-depth individual interviews (IDIs). The participants were community members, community healthcare workers, and Emergency Department healthcare workers. The study settings were rural areas of Karachi; Ibrahim Hyderi where the cluster randomized trial was conducted, and three hospitals, which were considered as the catchment population of the area.

### 2.2. Study Population and Setting

Focus group discussions were conducted with the residents of Ibrahim Hyderi (local vicinity in Karachi), including community healthcare workers. The sample pool of FGDs comprised a wide range of participants to ensure variability in the responses. They were approached through a health organization working in the area which is trusted by the community. This health organization was part of the Heat Trial. The FGDs were conducted at the office of the health organization. The discussion spanned around 45 min.

In-depth interviews were conducted with emergency care workers of three major hospitals in Karachi. All the interviews were conducted at the participant’s hospital, at a convenient place and time. The average duration of the interviews was 30 min.

### 2.3. Data Collection and Analysis

The FGDs and IDIs were conducted until saturation was achieved [26]. The research team trained in qualitative research, moderated the FGDs and IDIs. The research team and participants were unknown to each other to establish a common epistemological ground for the interviews [27]. The FGDs and IDIs were conducted by the two members of the research team (NA & RN). A semi structured IDI and FGD guide was prepared by the research team, which was reviewed by a qualitative research expert and a heat emergencies expert. (Appendix A and Appendix B). All IDIs and FGDs were conducted in the local language, Urdu, and were audio-recorded, after obtaining written consent from all the participants. Each participant was assigned a unique code. Confidentiality was maintained using these unique codes during transcription of the audio interviews. This allowed the researchers to anonymize the interviews while identifying valuable information like demographics, themes, and subcategories.

The interviews were transcribed by two authors (NA and RN). These authors further used content analysis approach [28] to identify the emerging themes and subthemes from the transcripts (Figure 1). Consensus was then reached between the two authors and a thematic dictionary was created, with definitions for each theme and quotes to support it. Rigor and trustworthiness in the study was established through following Lincoln and Guba’s criteria [29]. The study’s credibility was enhanced by emphasizing the purpose to learn from respondents through open and nonjudgmental attitude of interviewer during FGDs and IDIs.

Approval to conduct the study was obtained from the Ethics Review Committee of the Aga Khan University, Karachi, Pakistan.

## 3. Results

A total of 30 people participated in the FGDs (Table 1). The mean age of the sample was 34 years. The women were housewives and community healthcare workers whereas half the men were fishermen, and the rest were a tailor, students, and office workers. Among the participants, 60% had intermediate level education.

### 3.1. Themes Emerging from FGDs with Community Residents

Respondents discussed the challenges that the community face during periods of extreme heat in Karachi. In-depth narratives revealed the preventive and treatment mechanisms used by community to deal with heat emergencies. Two themes were drawn from the narratives depicting recognition and treatment of heat emergencies in the community and access to and quality of care in the hospitals in terms of heat crises.

#### 3.1.1. Recognition and Early Treatment of Heat Emergencies at Home

Participants from all the three FGDs admitted that early recognition of symptoms is important in managing patients with heat emergencies. One of the challenges in treating heat-related cases at home is that it is not recognized in the early stages of heat emergencies, which prevents the implementation of appropriate care. They expressed that cases are not recognized at the earlier stages of the condition, which is the biggest barrier in the treatment. Initially, the cases are treated for raised body temperature with acetaminophen, home remedies, and sprinkling water over the head and eyes if the patient has lost consciousness. The majority mentioned lack of financial resources as the cause of over reliance on home remedies, while others stated that people lacked the ability to recognize heat-induced conditions. A CHW from community informed,


*“People in the community keep treating patients at home at the initial stages, the decision to go to the hospital is taken in extreme conditions. Sometimes they call us, and we help them but most of the time treatment takes place at home with home remedies.”*
***(FGD # 3)***

Almost all the participants in the FGDs stated that drinking plenty of water during hot weather could help prevent heat emergencies. Moreover, the community residents informed that drinking lemonade and other homemade drinks, such as *“lassi”* (a mixture made of yogurt, milk, and water) helps hydrating the body. However, they emphasized that only simple water is enough to rehydrate if it is available. A male member of the community narrated,


*“Drinking lemonade has good effects on the body on hot summer days, I drink a lot whenever I am feeling low”*
***(FGD # 1)***

Similarly, a mother from the community verbalized,


*“I often make lemonade for my kids once they are back home from school, particularly, when I feel that weather is hot outside and my kids might have been affected by the weather”*
***(FGD #2)***

Community residents, both males and females perceived that drinking water or taking a shower soon after entering home in hot weather can cause paralysis of body or other infectious diseases and thought that this practice needed to be avoided. A male member from community verbalized,


*“I always advise my kids to avoid the practice of drinking cold water immediately after entering home in hot weather; take rest for a few minutes to allow the body to cool down. Drinking water immediately could cause other diseases, therefore, drink water once your body temperature comes to normal”*
***(FGD # 1)***

Modification in lifestyle can play a pivotal role in preventing heat emergencies such as drinking plenty of water and reducing strenuous activities. Participants narrated that eating less spicy foods and wearing light colored clothes help the body in maintaining its homeostasis. The male members thought that wearing a cap when going out in the sun and drinking water frequently during work can help a person stay safe from the harmful effects of heat. On the other hand, the community health workers (CHWs) believed that the community residents are not always aware of these preventive measures, which results in incidence of dehydration and heat emergencies. Both, the male community residents and CHWs felt that the structure of the houses in the community also contributed to heat emergencies, as the element of ventilation is rarely considered during construction. One male member from community expressed,


*“The design of the house is rarely considered during construction of the house, even the rooms within the house are not well ventilated”*
***(FGD # 1)***

Heat emergencies in the community can be considerably prevented by ensuring the adequate supply of water and electricity. In summers, the minimal availability of both largely affects health of community dwellers. Frequent power outrages for hours and sometimes for a stretch of days affect supply of water and, because of which, the community residents must drink stored and unhygienic water. A female from community narrated that,


*“We face a lot of issues with the supply of water in summers, unavailability of electricity hampers water supply. We bring water from our neighborhoods and store”*
***(FGD #2 group)***

#### 3.1.2. Access and Quality of Care in the Hospital

The number of healthcare facilities available in the community is inadequate with limited health services only targeting specific diseases. One male member of community stated that lack of transport facilities to transfer patients to healthcare facilities posed an added financial burden on the families. Therefore, often the decision regarding seeking healthcare depends on the financial resources, and medical care is accessed only in extreme situations.


*“We have to hire a private car to transfer our patients to a larger healthcare facility. We take this decision in cases when the patient is critically ill or about to die and when all the home remedies have been tried on the patient […] because we can’t afford private transport”*
***(FGD # 1)***

Moreover, the participants expressed concerns regarding lack of trust in emergency medical services. They stated that poor quality of services, harsh behavior of the healthcare professionals and the complex process of getting care at public sector hospitals made it difficult to avail healthcare services. They, therefore preferred home remedies or seeking healthcare from nearby private healthcare facilities, despite financial hardship. One participant said,


*“I have a very bad experience of going to a doctor because of number of reasons, one of them is their harsh behavior.”*
***(FGD #3)***

Cultural constraints are another contributing factor for not seeking healthcare, particularly among women because they are not allowed to go alone without a male chaperone. As one female verbalized,


*“We have to wait till evening for our males to arrive home and accompany us to a doctor and often the nearby clinics are closed by then”*
***(FGD #2)***

### 3.2. Themes Emerging from IDIs of Healthcare Professionals

The participants in the IDIs were doctors and nurses ranging from 25 to 50 years of age and all of them were involved in the management of patients during the heat wave emergency of 2015 (Table 2). Three themes emerged from the interviews with the healthcare professionals. However, both healthcare professionals and community members emphasized the importance of the early recognition and treatment of heat emergencies.

#### 3.2.1. Recognition and Treatment of Heat Emergencies

Generally, the healthcare providers felt confident of their ability to recognize heat illnesses. They thought that they were more aware of signs and symptoms of heat illnesses since the heat wave of 2015. When patients visit an emergency department with dehydration, dizziness and rapid pulse; healthcare providers recognize that this is heat illness and provide treatment to them accordingly. A male doctor from a public health facility stated,


*“So sometimes in extreme heat when the temperature rises up to 40 degrees, patients present with dehydration. They often visit with complaints about altered level of consciousness and shivering or chills as well. So, in this scenario we hydrate them for their survival.”*
***(IDI 01-DP20N-M)***

The respondents further explained other signs and symptoms that patients affected by heatwaves, such as lethargy, low blood pressure, rapid heart-beat, and high body core temperature. A male nurse from a public health facility stated,


*“Patients are lethargic, they have low blood pressure, and if they have had more sun exposure, they come with high temperature as well”*
***(IDI 02-NP7Y-M)***

Regarding treatment of patients with heat emergencies, healthcare providers bring down the patient’s temperature by sponging, icing, and keeping them in air-conditioned room. Further, they follow the workup for heat affected patients as soon as the patient is identified as having a heat stroke. Two participants expressed this as follows,


*“To manage heat stroke patients, we mainly do sponging. We have a tub and pipe for them to take showers, we do icing, we keep them in an airconditioned area and lastly we give 5% Dextrose, if not controlled.”*
***(IDI 06-DDY-M)***


*“Patients often do complain of severe headache as soon as they reach to hospital. We immediately take them inside and sponge their heads, we check their temperature, secure IV line, and hydrate them.”*
***(IDI 04-ND28Y-F)***

#### 3.2.2. Facility Level Plan

Based on their experience, the participants stated that dealing with heat emergencies required facility level planning and physical resources. They mentioned that essential supplies are needed to handle heat emergencies as per the plan. They also emphasized the need for a proper management plan to address the burden of heat emergencies; plans were being followed to some extent in each facility. A respondent from a public health facility stated,


*“We need ample fluids because dehydration is common in heat strokes. We should be prepared to have all the items that should be in the crash cart such as IV cannula, oxygen masks, medicines and intubation for more sick patients.”*
***(IDI 02-NP7Y-M)***

The participants further highlighted the need for a multidisciplinary team to manage heat-stroke. A doctor and a nurse from the public health facilities expressed their views as follows,


*“This should be teamwork, not only work of the emergency department. We have to involve other specialties such as medicine department, nephrology department, cardiology etc. […] Sometimes patients become very sick, and this leads to sepsis. So, in this situation an infectious disease specialist should be contacted to deal with such patients. So, teamwork is essential, without teamwork it is not possible.”*
***(IDI 01-DP20N-M)***


*“The first priority is that there should be air-conditioned areas so that patients get cooled directly as they enter the hospital, and their temperature stays on low. Simultaneously, fluids should be provided to them to balance their electrolytes”*
***(IDI 03-ND28-H-F)***

A doctor from the public hospital mentioned the physical resources required for heat stroke patients, such as proper ICU, ward, and ambulance for timely transportation. She further explained that a standard set of requirements is the same for routine patient and patients affected with heat-stroke. A female doctor from a public health facility mentioned,


*“Proper ICU is required, proper ward care is required, an ambulance to shift patients is required, so we need everything that a normal routine patient need”*
***(IDI 05-DD29-Y-F)***

#### 3.2.3. Training

Most of the participants emphasized that there should be regular trainings on the management of heat emergencies. The training should cover all the components, from basic to complex case scenarios, on disease recognition, diagnosis, and management. In addition, they emphasized on the availability of guidelines for uniformity in practices. They stated that guidelines will improve the clinical decision making of the healthcare providers. As two of the participants said,


*“There should be regular trainings and reinforcement on the implementation of guidelines to bring uniformity in practice.”*
***(IDI 03-ND28-H-F)***


*“There should be theoretical and practical training so that one is aware that summer is coming, and what should be the criteria to deal with heat stroke patients.”*
***(IDI 01-DP20N-M)***

## 4. Discussion

A qualitative approach was used in this study to investigate the perceptions and practices of a local Karachi community and its healthcare providers regarding recognizing and managing heat emergencies. Both the community and healthcare providers were aware of heat emergencies, especially after the 2015 Karachi heat wave. The discussion with community members revealed that the socioeconomic status of households can have a significant impact on the treatment of heat illnesses. The private sector of the healthcare system in Pakistan delivers 70% of the total healthcare services which is based on fee for services [30]. There is also a huge disparity in accessing healthcare services with 30% of the population living with absolute poverty. In addition, the healthcare expenditure as percentage of the gross domestic product is only 3.2% in Pakistan [31]. Similar findings have previously been observed; individuals from low socioeconomic status had poorer health outcomes and higher mortality because they lived in small, overcrowded housing with limited access to water, cooling appliances, and health facilities, as seen in Karachi’s 2015 heat wave [32,33,34]. Emergency care workers identified an increased burden on the emergency department as a result of heat illnesses in a city that already has a high prevalence of endemic illnesses [35].

Outdoor laborers were found to be more vulnerable to heat emergencies. These results are consistent with a study from Ahmedabad, India, which listed outdoor workers as a vulnerable group for heat illnesses [36]. Previous studies have identified workers in humid indoor/outdoor conditions [14] and women working in kitchens as vulnerable groups, which were not identified in our study. Heat emergencies have previously been identified as largely preventable through the provision of necessities such as water and electricity, as well as educating about the health risks of heat [15,21,22]. Furthermore, drinking traditional yoghurt, which is commonly used in South Asia provides the body with the liquid and nutrients in an easily digestible form, which are lost while sweating as an effect of exposure to heatwave. [37].

Members of the community identified financial and structural barriers to accessing emergency care for heat illnesses. In Pakistan, financial constraints are a common and consistent barrier to accessing healthcare during any illness, including heat emergencies [38]. Furthermore, cultural barriers, such as the belief that a woman visiting a healthcare facility without being accompanied by a male member is unacceptable, hinder timely treatment for heat-related illnesses, despite the fact that women are more vulnerable to heat emergencies while performing domestic chores. Cultural barriers are one of the many reasons for low utilization of healthcare services in many low- and middle-income countries (LMICs) [39]. Poor quality of emergency care services was perceived to be an impeding factor by the communities in obtaining healthcare for heat-related illnesses. Poor quality of care is one of many factors contributing to the underutilization of public healthcare facilities in LMICs [40].

Emergency department personnel had little knowledge of the signs and symptoms of heat illnesses, such as describing shivering as one of the symptoms, which is not indicative of heat illnesses, and mentioning the use of Dextrose 5% fluid as a treatment, which is not a standard of care [41]. In addition, differentiating between body core, skin, and air temperatures is critical for accurate diagnosis of heat emergencies. In Karachi, there is no uniform method for measuring body core temperatures in hospitals. However, measuring body core temperature and obtaining a history of heat exposure where possible are important for diagnosing heat-related illnesses, especially heat stroke, in the emergency department [42]. In the context of growing local diseases and outbreaks, matching the symptoms with heat was deemed challenging. Our results are consistent with a study conducted in Germany, in which general practitioners (GPs) were unable to identify environmental conditions as possible risk factors for heat emergencies [43].

The healthcare professionals expressed the need for a facility level response plan to deal with heat emergencies effectively. The heatwave in 2015, in Karachi, led to the development of the first heatwave management plan, in consultation with national and international experts [12]. This plan set out strategies for relevant agencies to ensure timely information on weather conditions, interagency coordination, and a public triggered activation system. In addition, strengthening primary healthcare services, to make them more responsive towards the management of the heat-related illnesses, can reduce the burden of heat emergencies in communities. Although ineffective primary care was not identified directly by our community, previous literature shows that the primary healthcare system serves as an early treatment hub for patients [44]. However, this is lacking in Pakistan [45]. Primary healthcare services may improve the health outcomes of the patients by detecting diseases in their early stages, and it can be one of the core elements in the implementation of an extreme heat disaster management plan [46].

### Limitations and Strength of the Study

This study had several limitations. Despite the small sample size, the potential themes emerged by grouping the responses of the participants and saturation in the responses was achieved. The practices reported by the emergency healthcare workers might have been influenced by the temptation to offer desirable responses believing their activities were being scrutinized. Moreover, the findings of the study are not intended to be generalizable beyond the scope of the study. Despite these limitations, the study provided contextual knowledge to facilitate the construction of heat prevention and treatment guidelines for communities and healthcare professionals. The study explored the perceptions of the community and healthcare workers simultaneously, which is a strength of this study. The findings from this study can be applicable to other resource constrained settings similar to Karachi. The heterogeneity in the sample pool was ensured by recruiting a wide range of male and female respondents including community dwellers, nurses, physicians and administrative workers who were knowledgeable about and had experience of a phenomenon of interest [47]. Perspectives of policy makers and representatives of the implementation agencies could be another interesting aspect to include in future research.

## 5. Conclusions

This qualitative study has provided new insights into the context of communities and healthcare professionals facing consequences of heat exposure in Karachi. Study findings suggest that there is an awareness about heat emergencies among community members in Karachi. The community perceived dehydration as a heat emergency and managed it with home remedies, cooling, and hydration. Furthermore, heat emergencies were identified to be connected with a shortage of electricity and water supply. The healthcare workers had limited awareness about the signs and symptoms of heat emergencies and perceived the early recognition and treatment of heat emergencies as challenging, in the face of endemic infections having similar presentation. Furthermore, poor quality of public healthcare services, inadequate training, and ineffective implementation of heat wave preparedness plans were identified as impeding factors in the treatment heat emergencies. Considering these aspects, there is a need to carry out preventive actions that take into account the socioeconomic challenges of the communities. This may inform heat prevention policies in communities facing longer and more intense hot spells.

## Figures and Tables

**Figure 1 ijerph-18-04736-f001:**
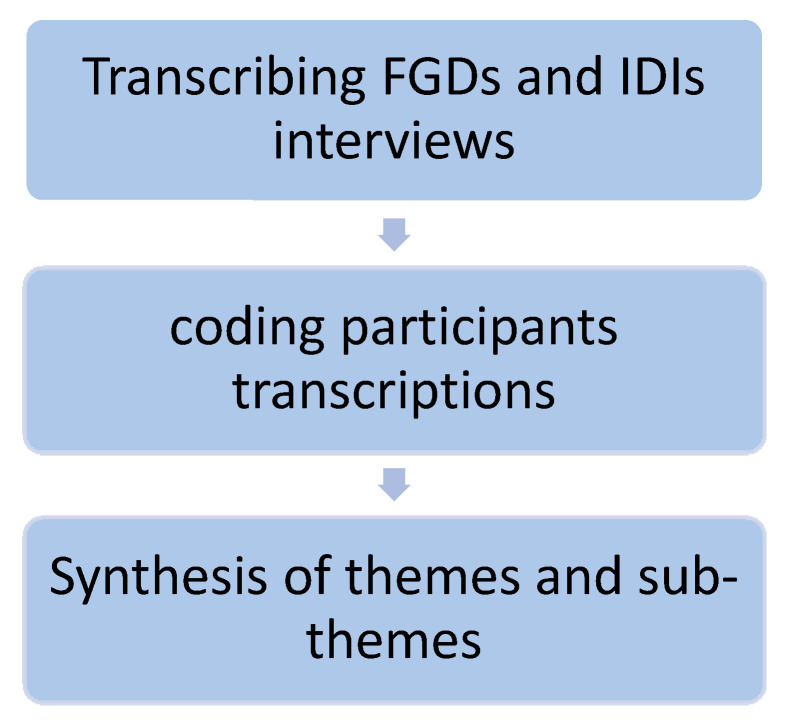
Qualitative data analysis sequence.

**Table 1 ijerph-18-04736-t001:** Demographic characteristics of the participants from FGDs.

Characteristics	Total (*n* = 30)
Age (mean, years)	34
Gender	
Male	12 (40%)
Female	18 (60%)
Occupation	
Housewife	10 (33.3%)
Fisherman	5 (16.6%)
Tailor	1(3.3%)
Community health workers	10 (33.3%)
Office work	2 (6.6%)
Student	2 (6.6%)
Formal education	
Grade 10 and below	5 (16%)
10–12 grades	18 (60%)
>12 years	1 (3%)

**Table 2 ijerph-18-04736-t002:** Demographic characteristics of the IDI participants.

Characteristics	Total (*n* = 6)
Age (mean, years)	38
Male	3 (50%)
Female	3 (50%)
Profession	
Doctor	3 (50%)
Nurse	3 (50%)
Work experience (range, years)	5 to 25
Experience of 2015 heat wave	6 (100%)

## Data Availability

The data presented in this study are available on request from the corresponding author.

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
