# Peer review of "Heat Emergencies: Perceptions and Practices of Community Members and Emergency Department Healthcare Providers in Karachi, Pakistan: A Qualitative Study"

_ijerph, 2021, doi:10.3390/ijerph18094736_

Round 1

Reviewer 1 Report

Dear authors, thank you for giving me the possibility to revise this interesting qualitative study describing perceptions and practices related to heat emergencies.

Although I found the paper well-structured, methodologically sound, and leading to acceptable implications for practice and healthcare policies, there are some rooms in which your manuscript should be improved.

Firstly, I would recommend accurate proofreading of your paper to improve the use of the English language and, subsequently, the manuscript's readability.

Please, follow my comments below, in which I suggest how to improve your paper. 

Introduction: 

Page 1, line 28 "emergencies are a public health problem."

Page 1-2, lines 37-44 I found this paragraph very hard to follow. Please, consider rewriting it more narratively. It seems like a list of years, Countries, and percentages, affecting the readability of the sentences.

Methods:

Page 2, lines 78-79 Please, clarify how this qualitative part of the study is embedded/linked to the HEAT trial.

Page 2, line 81 Please, pay attention to punctuation.

Page 2, line 85 I would found better if more details about how, when and where FG/IDI were performed.

Page 3, lines 95-97 What was the relationship between interviewers/researchers and interviewees/participants? Where their colleagues? Did they know them?

Page 3, lines 107-109 Where the FGs and IDIs analyzed using the same procedure? Please, describe in-depth the phases of the thematic analysis you performed, who performed such phases, and what strategies have been put in practice to ensure trustworthiness.

Results:

Tables: why did you express the mean age for table 1 and range in table 2? I would suggest you be consistent in reporting data.

Page 3, lines 119 I would suggest you write a small introduction/presentation describing the themes that emerged from your data and the patterns you can find among themes.

Page 5, line 194 If there is a long pause between participants' words, put three dots ... , if you have opted to report two different sentences omitting an irrelevant part, put three dots in square brackets [...].

Page 6, line 257 I wondered if it is necessary to report the title/occupation of participants. My suggestion is to omit any characterizing detail from a qualitative study to limit the likelihood of recognizing participants.

Discussion

Your results have stated that the lack of financial resources may cause over-reliance on home remedies. Well, I would suggest you include a brief description of the Pakistani healthcare system, as it can improve comprehensibility on the reasons for those limited financial resources.

Moreover, I would suggest you write the strength and limitations section at the end of the discussion. You should report the many limits typical of qualitative study and those specific to your context and the strategies you have implemented to reduce such limits.

Conclusions:

Please, include a/more sentences on possible solutions you suggest, coherently with your findings.

I hope these suggestions will help you improve your manuscript, looking forward to examining the revised version.

Author Response

Manuscript ID

ijerph-1151540

Title

Heat Emergencies: perceptions and practices of Emergency Department healthcare providers and community members in Karachi, Pakistan: a qualitative study

Authors

Uzma Rahim Khan * , Naveed Ahmad , Rubaba Naeem , Umerdad Khudadad , Sarwat Masud , Nadeem Ullah Khan , Junaid Razzak

External Peer-Review Report

Reviewer 1 Comments

Comments

Action Taken

Dear authors, thank you for giving me the possibility to revise this interesting qualitative study describing perceptions and practices related to heat emergencies.

Thank you for your reviewing our manuscript to improve it further.  

Although I found the paper well-structured, methodologically sound, and leading to acceptable implications for practice and healthcare policies, there are some rooms in which your manuscript should be improved.

Thank you for the encouraging comment.

Firstly, I would recommend accurate proofreading of your paper to improve the use of the English language and, subsequently, the manuscript's readability.

The language of the paper was revised by a professional English instructor.

Please, follow my comments below, in which I suggest how to improve your paper. 

Thank you for the valuable feedback.

Introduction: 

Page 1, line 28 "emergencies are a public health problem."

Addressed accordingly.

Page 1-2, lines 37-44 I found this paragraph very hard to follow. Please, consider rewriting it more narratively. It seems like a list of years, Countries, and percentages, affecting the readability of the sentences.

Rewrote the paragraph as suggested. Please see Page 1-2, lines 38-48. The UNHCR's report on weather-related incidents from 1995 to 2015 highlighted the im-portance of risk identification and reduction in vulnerable countries. With regard to heat events, Chicago, United States reported 514 deaths in July 1995 followed by California, that recorded a higher number of emergency visits in 2006. Similar burden of heat emergencies heat has been observed in Western Europe including France, Germany and Spain.  In addition, New South Wales reported a 13% rise increase in all-cause mortality, and a 14% increase in ambulance calls related to heat emergencies in 2011. Furthermore, India reported 1,000 excessive deaths in Ahmedabad as a result of extreme heat in 2010.

Methods:

Page 2, lines 78-79 Please, clarify how this qualitative part of the study is embedded/linked to the HEAT trial.

Addressed! Please see page 2, lines 83-87.

One of the objectives in the heat trial was to develop a community heat prevention guideline. To facilitate the contextual development of the guideline catering to the needs of community and emergency healthcare workers, a qualitative study was conducted using focus group conversations (FGDs) and in-depth individual interviews (IDIs).

Page 2, line 81 Please, pay attention to punctuation.

Addressed! Colon removed.

Page 2, line 85 I would found better if more details about how, when and where FG/IDI were performed.

Addressed! Please see page 3, lines 90-99.

Focus group discussions were conducted with the residents of Ibrahim Hyderi (local vicinity in Karachi), including community health care workers. The sample pool of FGDs comprised of wide range of participants to ensure variability in the responses. They were approached through a health organization working in the area which is trusted by the community. This health organization was part of the Heat Trial. The interviews FGDs were conducted at the office of the health organization. The discussion spanned around 45 minutes.

In-depth interviews were conducted with emergency care workers of three major hospitals in Karachi. All the interviews were conducted at the participant’s hospital, at a convenient place and time. The average duration of the interviews was 30 minutes.

Page 3, lines 95-97 What was the relationship between interviewers/researchers and interviewees/participants? Where their colleagues? Did they know them?

Addressed! Please see page 3, lines 106-108.

The research team and participants were unknown to each other to establish a common epistemological ground for the interviews

Page 3, lines 107-109 Where the FGs and IDIs analyzed using the same procedure? Please, describe in-depth the phases of the thematic analysis you performed, who performed such phases, and what strategies have been put in practice to ensure trustworthiness.

Addressed! Please see page 3, lines 114-119. Both, FGDs and IDIs were independently interpretated by two authors (NA & RN) using the similar content analysis approach. Rigor and trustworthiness in the study was established through following Lincoln and Guba’s criteria. The study's credibility was enhanced by emphasizing the purpose to learn from respondents through open and non-judgmental attitude of interviewer during FGDs and IDIs.

Results:

Tables: why did you express the mean age for table 1 and range in table 2? I would suggest you be consistent in reporting data.

Addressed! Range is used in both tables.

Page 3, lines 119 I would suggest you write a small introduction/presentation describing the themes that emerged from your data and the patterns you can find among themes.

Addressed! Please see page 4, lines 139-143.

Respondents discussed the challenges that community face during periods of extreme heat in Karachi. In-depth narratives revealed the preventive and treatment mechanisms used by community to deal with heat emergencies. Two themes were drawn from the narratives depicting recognition and treatment of heat emergencies in the community and access to and quality of care in the hospitals in terms of heat crises.

Page 5, line 194 If there is a long pause between participants' words, put three dots ... , if you have opted to report two different sentences omitting an irrelevant part, put three dots in square brackets [...].

Addressed accordingly Please see page 5, line 216.

Page 6, line 257 I wondered if it is necessary to report the title/occupation of participants. My suggestion is to omit any characterizing detail from a qualitative study to limit the likelihood of recognizing participants.

Addressed accordingly. Please see page 7, line 280

Discussion

Your results have stated that the lack of financial resources may cause over-reliance on home remedies. Well, I would suggest you include a brief description of the Pakistani healthcare system, as it can improve comprehensibility on the reasons for those limited financial resources.

Addressed accordingly! Please see page 8, line 331-335.

The private sector of the healthcare system in Pakistan delivers 70% of the total healthcare services which is based on fee for services. There is also a huge disparity in accessing healthcare services with 30% of the population living with absolute poverty. In addition, the healthcare expenditure as percentage of the gross domestic product is only 3.2 percent in Pakistan

Moreover, I would suggest you write the strength and limitations section at the end of the discussion. You should report the many limits typical of qualitative study and those specific to your context and the strategies you have implemented to reduce such limits.

Addressed! Please see page 9, lines 389-403.

This study had several limitations. The sample sizes for IDIs and FGDs were relatively small. However, this issue was mitigated by grouping the participants responses around the themes presented and no response was omitted from being presented in the results. The practices reported by the emergency healthcare workers might have been influenced by the temptation to offer desirable responses believing their activities were being scrutinized. Moreover, the findings of the study are not intended to be generalizable beyond the scope of the study. Despite these limitations, the study provided contextual knowledge to facilitate the construction of heat prevention and treatment guidelines for communities and healthcare professionals.

The study explored the perceptions of the community and healthcare workers simultaneously, which is a strength of this study. The findings from this study can be applicable to other resource constrained settings similar to Karachi. The heterogeneity in the sample pool was ensured by recruiting male and female from both the local community of people and the health care workers. Perspectives of policy makers and representatives of the implementation agencies could be another interesting aspect to include in future research.

Conclusions:

Please, include a/more sentences on possible solutions you suggest, coherently with your findings.

I hope these suggestions will help you improve your manuscript, looking forward to examining the revised version.

Thank you for the valuable comment. Please see page 10, lines 419-435.

This qualitative study has provided new insights into the context of communities and healthcare professionals facing consequences of heat exposure in Karachi. Study findings suggest that there is an awareness about heat emergencies among community members in Karachi. The community perceived dehydration as a heat emergency and managed it with home remedies, cooling, and hydration. Furthermore, heat emergencies were identified to be connected with a shortage of electricity and water supply. The healthcare workers had limited awareness about the signs and symptoms of heat emergencies and perceived the early recognition and treatment of heat emergencies as challenging, in the face of endemic infections having similar presentation. Furthermore, poor quality of public health care services, inadequate training, and ineffective implementation of heat wave preparedness plans were identified as impeding factors in the treatment heat emergencies. Considering these aspects, there is a need to carry out preventive actions that take into account the socio-economic challenges of the communities. This may inform heat prevention policies in communities facing longer and more intense hot spells.

Reviewer 2 Report

Estimated Authors,

Estimated Editors,

thank you for the opportunity to review this very interesting paper on perceptions and practices of Emergency Department healthcare providers and community members in Karachi, Pakistan.

This paper represents a qualitative study, ie a study that tentatively identify and describe topics that would be explored through quantitative and/or semiquantitative studies on the very same populations. In other words, being a sort of spark for future researches, potential limits represented by limited homogeneity of the results, reduced sampling size, etc not necessarily represent a main concern.

In facts, the present paper may be eventually accepted for publication after a couple of adjustments, and namely:

  1. As the text initially reports on community members and then on HCP, I would invert the topics in the main title (i.e. "Heat Emergencies : perceptions and practices of community members and Emergency Department healthcare providers in Karachi , Pakistan : a qualitative study");
  2. Tables should include (where possible) percent values.
  3. Please provide further details on how and when the participants were eventually selected.

Author Response

Manuscript ID

ijerph-1151540

Title

Heat Emergencies: perceptions and practices of Emergency Department healthcare providers and community members in Karachi, Pakistan: a qualitative study

Authors

Uzma Rahim Khan * , Naveed Ahmad , Rubaba Naeem , Umerdad Khudadad , Sarwat Masud , Nadeem Ullah Khan , Junaid Razzak

External Peer Review Report

Reviewer 2 Comments

thank you for the opportunity to review this very interesting paper on perceptions and practices of Emergency Department healthcare providers and community members in Karachi, Pakistan.

Thank you for reviewing our paper and providing constructive feedback.  

This paper represents a qualitative study, ie a study that tentatively identify and describe topics that would be explored through quantitative and/or semiquantitative studies on the very same populations. In other words, being a sort of spark for future researches, potential limits represented by limited homogeneity of the results, reduced sampling size, etc not necessarily represent a main concern.

In facts, the present paper may be eventually accepted for publication after a couple of adjustments, and namely:

  1. As the text initially reports on community members and then on HCP, I would invert the topics in the main title (i.e. "Heat Emergencies : perceptions and practices of community members and Emergency Department healthcare providers in Karachi , Pakistan : a qualitative study");

Thank you for the feedback. We second to your remark on the limitations of the qualitative study. However, this study was intended to facilitate the construction of contextual heat prevention guidelines for the communities and health facilities.

Thank you for refining our title of the study to a more meaningful one.

2. Tables should include (where possible) percent values.

Addressed accordingly! Percent values added in both the tables where it was applicable.

3. Please provide further details on how and when the participants were eventually selected.

Addressed! Please see page 3, lines 90-99.

Focus group discussions were conducted with the residents of Ibrahim Hyderi (local vicinity in Karachi), including community health care workers. The sample pool of FGDs comprised of wide range of participants to ensure variability in the responses. They were approached through a health organization working in the area which is trusted by the community. This health organization was part of the Heat Trial. The interviews FGDs were conducted at the office of the health organization. The discussion spanned around 45 minutes.

In-depth interviews were conducted with emergency care workers of three major hospitals in Karachi. All the interviews were conducted at the participant’s hospital, at a convenient place and time. The average duration of the interviews was 30 minutes

Reviewer 3 Report

The article addresses important issues concerning the effects of climate change i.e. prolonged periods of high temperature. The researchers relied on interviews among different social groups. This approach to the problem seems appropriate and could provide reliable data. Unfortunately, the group of respondents seems to be too modest, as it consists of only 30 people in the focus group and 6 among the medical staff. It should be mentioned that Karachi has a population of 15 million. This makes the results presented here preliminary and highly causal research rather than an in-depth analysis of the problem. I am afraid that in such a research group the randomness of the results is too high. The conclusions are very limited. 
As far as the technical side of the research is concerned, the results are not very interesting. The authors quoted selected answers of the respondents. How is this supposed to interest a potential reader? It seems that the answers of patients and medical personnel should have been collated and some sort of gradation of health problems or/and counteracting the effects of severe heat should have been made. Overall, the results are presented enigmatically without any specific survey results.
It is not described how deaths were related to heat for example, to patients' diseases which were intensified by high temperatures.
It is puzzling that there is no comparison of climate parameters (temperature, humidity) in the study period with other years (last 10-15 years). This would show the real problem faced by the people of Karachi. 
Specific comments (selected):
The summary is generally constructed correctly but I would nevertheless supplement it with some "hard" research data.
line 28 is a very simplistic definition of global climate change. High temperature is one of several symptoms of climate change. In other parts of the world, violent snowstorms or tornadoes etc. may be a sign of change.
line 32 the percentage given confirms the previous remark
line 32-34 The sentence is incorrectly written. There is no difference, factors similarly affect regardless of the wealth of countries
 line 43-44 The same way of stating data as in the previous sentence should be retained.  It is difficult to deduce whether the mortality with India was higher, lower because there is no point of reference 
line 45-46 which factors are referred to in this sentence
line 47 very general statement - it is difficult to identify authors intentions
Most of the author's statements are very obvious e.g. line 136 It is hard to expect lack of knowledge about the necessity to drink water in hot weather. 
line 139 "refresh the body" - what does this mean? Is it about hydrating the body?
line 374 the sentence is unclear, besides from what does this conclusion follow.
The discussion refers to the results obtained from the study. This part of the study deals with more or less obvious statements i.e. heat is more dangerous for people outside buildings, pointing out sick people and children as a particularly vulnerable group ect. I am unable to find any clearly new information that is relevant to the development of knowledge about the effects of heat. 
Lines 361-370 attempt to indicate the strength of the study. Unfortunately, although the topic is extremely important, the way the results are presented is insufficient. I am sorry, but I do not share the authors' opinion in this regard.

Author Response

Manuscript ID

ijerph-1151540

Title

Heat Emergencies: perceptions and practices of Emergency Department healthcare providers and community members in Karachi, Pakistan: a qualitative study

Authors

Uzma Rahim Khan * , Naveed Ahmad , Rubaba Naeem , Umerdad Khudadad , Sarwat Masud , Nadeem Ullah Khan , Junaid Razzak

External Peer Review Report

Reviewer 3 Comments

The article addresses important issues concerning the effects of climate change i.e. prolonged periods of high temperature. The researchers relied on interviews among different social groups. This approach to the problem seems appropriate and could provide reliable data. Unfortunately, the group of respondents seems to be too modest, as it consists of only 30 people in the focus group and 6 among the medical staff. It should be mentioned that Karachi has a population of 15 million. This makes the results presented here preliminary and highly causal research rather than an in-depth analysis of the problem. I am afraid that in such a research group the randomness of the results is too high. The conclusions are very limited. 

Thank you for the valuable comments. The perception of the respondents from the catchment population was explored based on the feasibility of the study using purposive recruitment of participants. In addition, the catchment population was intended to receive the intervention which projected the need to explore their perceptions. Generalizability was not the claim. Objective of the study was limited in scope to link the perceptions of the respondents with the development of guidelines for heat prevention and treatment.

As far as the technical side of the research is concerned, the results are not very interesting. The authors quoted selected answers of the respondents. How is this supposed to interest a potential reader? It seems that the answers of patients and medical personnel should have been collated and some sort of gradation of health problems or/and counteracting the effects of severe heat should have been made. Overall, the results are presented enigmatically without any specific survey results.

Thank you for the constructive feedback. However, the findings could not be combined since the participants in the FGDs and IDIs were heterogenous. For interviews, we used two different guides: one for community members (FGDs) and another for healthcare workers serving in the emergency rooms of public hospitals (IDIs) in Karachi. Furthermore, no point of association was found in the patterns of responses emerged from the FGDs and IDIs that should have been combined.

It is not described how deaths were related to heat for example, to patients' diseases which were intensified by high temperatures.
It is puzzling that there is no comparison of climate parameters (temperature, humidity) in the study period with other years (last 10-15 years). This would show the real problem faced by the people of Karachi. 

Addressed! Please see page 2 line 56-65.

Pakistan ranks 7th on the list of the top ten countries with the climate risk index score of 30.5 and a total number of 141 weather-related events in the past 10 years. Temperatures in Karachi soar to an extreme of 45°C during the months of May to July, along with high humidity 30 to 95% as a daily average from 2006 to 2015. The 30-year average maximum temperature for June, the hottest summer month in Karachi, is 34.8°C.

Specific comments (selected):
The summary is generally constructed correctly but I would nevertheless supplement it with some "hard" research data.

Thank you for your feedback. The findings of the paper is supported by the data obtained through FGDs and IDIs and none of the responses of the participants were omitted.

line 28 is a very simplistic definition of global climate change. High temperature is one of several symptoms of climate change. In other parts of the world, violent snowstorms or tornadoes etc. may be a sign of change.

Addressed! Please see page 1, lines 28-30

Global climate change has been related to rising temperatures, which have resulted in heat waves, more severe droughts, heavy rains, and intense hurricanes in various parts of the world.

line 32 the percentage given confirms the previous remark

Kindly specify this comment for better comprehension.

line 32-34 The sentence is incorrectly written. There is no difference, factors similarly affect regardless of the wealth of countries

Addressed! Please see page 1, lines 33-37.

Extreme air temperature contributes directly to cardiovascular and respiratory disease deaths, especially among the elderly. More than 166, 000 people died as a result of heatwaves between 1998 and 2017, including more than 70, 000 in Europe during the 2003 heatwave. Between 2000 and 2016, the number of people exposed to heatwaves went up by nearly 125 million.

 line 43-44 The same way of stating data as in the previous sentence should be retained.  It is difficult to deduce whether the mortality with India was higher, lower because there is no point of reference 

Addressed! Please see page 2, lines 53-54.

line 45-46 which factors are referred to in this sentence

Addressed! Please see page 2, lines 56-57

Pakistan ranks 7th on the list of the top ten countries with the climate risk index score of 30.5 and a total number of 141 weather-related events in the past 10 years.  

line 47 very general statement - it is difficult to identify

authors intentions

Addressed! The statement is revised. Please see page 2, line 58.

Most of the author's statements are very obvious e.g. line 136 It is hard to expect lack of knowledge about the necessity to drink water in hot weather. 

Agreed! The statement is quite general and removed.

line 139 "refresh the body" - what does this mean? Is it about hydrating the body?

Addressed! Please see page 4 line 161.

Moreover, the community residents informed that drinking lemonade and other home-made drinks such as, “lassi” (a mixture made of yogurt, milk, and water) helps hydrating the body.

line 374 the sentence is unclear, besides from what does this conclusion follow.

Addressed! The statement is revised. Please see page 9 lines 418-419.

The discussion refers to the results obtained from the study. This part of the study deals with more or less obvious statements i.e. heat is more dangerous for people outside buildings, pointing out sick people and children as a particularly vulnerable group ect. I am unable to find any clearly new information that is relevant to the development of knowledge about the effects of heat. 

Thank you for the feedback. However, it was a qualitative study with limited scope. The study findings were used to develop the contextual heat prevention and treatment guidelines.

Lines 361-370 attempt to indicate the strength of the study. Unfortunately, although the topic is extremely important, the way the results are presented is insufficient. I am sorry, but I do not share the authors' opinion in this regard.

We understand that qualitative study comes with various limitations and the scope of study remains narrow. Please see page 9, lines 389-403.

This study had several limitations. The sample sizes for IDIs and FGDs were relatively small. However, this issue was mitigated by grouping the participants responses around the themes presented and no response was omitted from being presented in the results. The practices reported by the emergency healthcare workers might have been influenced by the temptation to offer desirable responses believing their activities were being scrutinized. Moreover, the findings of the study are not intended to be generalizable beyond the scope of the study. Despite these limitations, the study provided contextual knowledge to facilitate the construction of heat prevention and treatment guidelines for communities and healthcare professionals.

The study explored the perceptions of the community and healthcare workers simultaneously, which is a strength of this study. The findings from this study can be applicable to other resource constrained settings similar to Karachi. The heterogeneity in the sample pool was ensured by recruiting male and female from both the local community of people and the health care workers. Perspectives of policy makers and representatives of the implementation agencies could be another interesting aspect to include in future research.

Round 2

Reviewer 1 Report

Dear authors, 

Thank you for allowing me to review the second version of this interesting paper.

I appreciated your responses to my comments, that were all considered. I would recommend to reduce the words of your introduction and discussion sections. Please, further consider to include a reference for the data saturation in the methods. Moreover, I would recommend a further revision of the English language.

Regards

Author Response

Manuscript ID

ijerph-1151540

Title

Heat Emergencies: perceptions and practices of Emergency Department healthcare providers and community members in Karachi, Pakistan: a qualitative study

Authors

Uzma Rahim Khan * , Naveed Ahmed , Rubaba Naeem , Umerdad Khudadad , Sarwat Masud , Nadeem Ullah Khan , Junaid Razzak

Reviewer 1 Comments

Thank you for allowing me to review the second version of this interesting paper.

I appreciated your responses to my comments, that were all considered. I would recommend to reduce the words of your introduction and discussion sections. Please, further consider to include a reference for the data saturation in the methods. Moreover, I would recommend a further revision of the English language.

Thank you for the valuable feedback.

Reference for the data saturation in the method section is added. Please see page 3, line 118. 26. (Fusch, P.I. and L.R. Ness, Are we there yet? Data saturation in qualitative research. The qualitative report, 2015. 20(9): p. 1408.)

The words in the introduction section reduced from 795 to 626 words.

The words in the discussion section reduced from 1141 to 1005 words.

The language of the manuscript has been improved further where needed.

Reviewer 3 Report

Thank you very much for the answers provided. I appreciate the effort of the authors in making corrections to the manuscript. It seems to me that the original problem of the small number of respondents is a key element. I do not think that the heterogeneity to which the authors refer is sufficient and eliminates the indicated weak point of the article. Maintaining the structure of the paper in the form of citations of respondents' answers is not very interesting. I was not persuaded that the paper can be of interest to a wide range of international scienters.   

Author Response

External Peer-Review Report

Reviewer Comments

Action Taken

Thank you very much for the answers provided. I appreciate the effort of the authors in making corrections to the manuscript. It seems to me that the original problem of the small number of respondents is a key element. I do not think that the heterogeneity to which the authors refer is sufficient and eliminates the indicated weak point of the article. Maintaining the structure of the paper in the form of citations of respondents' answers is not very interesting. I was not persuaded that the paper can be of interest to a wide range of international scienters.

Thank you for the valuable feedback. We understand that sample of respondents was relatively small, however saturation in the responses was achieved which is more important to determine sample size for qualitative studies. With reference to the heterogeneity in the characteristics of the respondents, please see page 10, lines 423-427. The heterogeneity in the sample pool was ensured by recruiting wide range of male and female respondents including community dwellers, nurses, physicians and administrative workers who were knowledgeable about and had experience of a phenomenon of interest. The citation of respondents’ answers is the standard reporting practice in qualitative studies.

This manuscript is a resubmission of an earlier submission. The following is a list of the peer review reports and author responses from that submission.